# A Critical Review of Smart City Frameworks: New Criteria to Consider When Building Smart City Framework

**Fan Shi**  **and Wenzhong Shi ***

Department of Land Surveying and Geo-Informatics, Otto Poon Charitable Foundation Smart Cities Research Institute, The Hong Kong Polytechnic University, Hong Kong 999077, China; 21041446r@connect.polyu.hk
* Correspondence: john.wz.shi@polyu.edu.hk

**Abstract:** In the face of persistent challenges posed by urbanization and climate change, the contemporary era has witnessed a growing urgency for urban intelligence and sustainable development. Consequently, a plethora of smart city schedules and policies have emerged, with smart city assessment serving as a pivotal benchmark for gauging policy effectiveness. However, owing to the inherent ambiguity of the smart city definition and the complexity of application scenarios, designers and decision-makers often struggle to ascertain their desired assessment frameworks swiftly and effectively. In this context, our study undertook a comprehensive analysis and comparative assessment of 33 recently introduced or inferred evaluation frameworks, drawn from a broad spectrum of extensive and longstanding research efforts. The overarching goal was to provide valuable reference points for designers and decision-makers navigating this intricate landscape. The assessment was conducted across seven key dimensions: generalizability, comprehensiveness, availability, flexibility, scientific rigor, transparency, and interpretability. These criteria hold the potential not only to guide the development trajectory and focus of upcoming smart city assessment models but also to serve as invaluable guidelines for stakeholders evaluating the outcomes of such models. Furthermore, they can serve as robust support for designers and decision-makers in their pursuit of targeted frameworks.

**Keywords:** smart city framework; critical analysis; sustainable development; literature review

## 1. Introduction

Urbanization represents a cohesive and worldwide trajectory of development and aspiration. Scholars project a scenario in which the global urban population will double by 2050, with approximately 70 percent of the global populace residing in urban regions [1]. Cities serve as hubs of diverse human activities, encompassing political, economic, and everyday life endeavors. Consequently, cities are expected to embody qualities of sustainability, efficiency, and inclusivity. However, the management and governance of urban centers often encounter significant challenges due to their intricate nature as multifaceted systems intertwined with various functions and intricate human-environment interactions. The necessity to formulate strategies for orchestrating developmental plans, addressing developmental complexities, and evaluating developmental benchmarks is of paramount importance.

The concept of the smart city was initially introduced in 1990 with the primary aim of integrating advanced information and communication technology (ICT) into urban planning [2]. As the concept of Smart Cities evolved to align with the needs of decision-makers and urban development, it became intertwined with various other notions such as intelligence, ubiquity, knowledge, information, and digitalization [3]. Essentially, the implementation of smart cities aims to enhance the transparency, accountability, effectiveness, and efficiency of interactions between citizens and government bodies through the incorporation of ICT technologies [3]. While the overarching goal of nearly all smart

city initiatives remains consistent—to create sustainable [4], efficient, and livable urban environments for residents—the proposed frameworks for smart city development can significantly differ due to varying interpretations of the smart city concept. Undoubtedly, given the distinct attributes of cities and the unique characteristics of nations, achieving a universally applicable framework for smart city development is challenging; nevertheless, overarching principles and standards for evaluating smart city endeavors should exist. This would facilitate the sharing of information and knowledge among diverse urban centers, ultimately guiding the trajectory of smart city development.

Within the previous research of smart city evaluation frameworks, divergent research focused on smart city literature reviews have manifested. One approach originates from the exploration of smart city research definitions to discuss the construction of smart city evaluation systems. Various measurement frameworks and indices have been developed to reflect different concepts of smart cities, including internet/digital cities, sustainable/green cities, knowledge/smart cities, etc. [5,6]. Thajba Aljowder et al. [7] embarked on this avenue, drawing upon the smart city definition summarized by ITU-T FGSSC: the outcome of the conducted analysis defines the smart city as "an innovative city that uses ICTs and other means to improve quality of life, the efficiency of urban operations and services and competitiveness while ensuring that it meets the needs of present and future generations with respect to economic, social and environmental aspects" [8]. They proceeded to investigate relevant smart city evaluation frameworks, evaluating the core contents of the studied models, and raising concerns about the disparities in framework applicability. Arroub [9] delved into the architecture and infrastructure of the framework, guided by the definition of a smart city encompassing intelligent economy, intelligent environment, intelligent mobility, intelligent living, and intelligent human capabilities. Notably, it was asserted that smart city development should be characterized as three-dimensional and reliant on the progression of information technology.

Another approach revolves around exploring research themes within the realm of smart cities, often prognosticating their future evolution. Parul Gupta provided a perceptive perspective on prevalent smart city assessment themes, underscoring the imbalanced distribution of indicators [3], with a predominant emphasis on novel technologies (such as supply chain technology and development) at the expense of social impacts, governance, and policy considerations. Yin C. et al. briefly reviewed diverse smart city frameworks, categorizing them based on their definition and application for the reference of fellow researchers [10]. Their categorization encompassed technical infrastructure, application domains, system integration, and data processing, asserting that smart city development should be contemplated through these four dimensions. Similarly, Andrés et al. confined the concept of smart cities to information cities, dissecting them as urban developments centered around citizen progress, open data, and sustainability, advocating for evaluation from these facets [11].

Yet another strand of research initiates discussions by probing into the standards of smart city evaluation systems. Ayyoob Sharifi also reviewed the most current smart city assessment tools and delineated 11 criteria for framework design [12]. This paper likewise embarks on discourse from this standpoint.

Smart city assessment represents an emerging field with significant growth potential. Consequently, researching and evaluating existing smart city frameworks and formulating standardized evaluation criteria hold profound significance [12]. Smart city assessment tools play a crucial role in accurately evaluating the implementation of smart cities, offering benefits to all diverse stakeholders involved [13]. Researchers and users of smart city assessment are searching for better assessment frameworks [14]. A comprehensive understanding and shared overarching principles governing the construction of smart city evaluations will profoundly influence the future development and planning of smart cities. Similar to the prevailing trend of economic globalization, the development of smart cities is gradually converging towards common future objectives. These shared foundational

principles will not impinge on the specific rules governing smart city development in individual nations, but they can provide overarching guidance and refinement.

Prior research indicates that diverse outcomes have arisen in the study of smart city frameworks due to disparities in research perspectives. Given the inherent ambiguity of smart city definitions, both their essence and projected developmental outcomes lack uniformity. This has posed substantial challenges for users seeking to locate and compare frameworks aligning with their specific needs. Hence, comparing and summarizing existing frameworks from distinct criteria not only aids users in swiftly identifying the requisite frameworks but also offers broader reference points for framework designers. Currently, the predominant approach in smart city research originates from the definition of smart cities, which leads to the development of smart city assessment frameworks based on individual designers' unique understanding of smart cities. This situation presents challenges for researchers or users to quickly identify their desired frameworks, preventing them from conducting comparisons and analyses from a relatively unified perspective. As a result, a comprehensive and holistic comparative system holds significant importance for the study of smart cities.

The purpose of this article is to introduce new principles that can serve as references for the construction of smart city evaluation frameworks, and subsequently analyze and compare 33 frameworks based on these new principles. These principles are derived from a synthesis and compilation of previous research. The results reveal that due to the diverse roles of framework designers, the varying purposes of the frameworks, and the geographical regions of the designers, the performance of frameworks differs across different principles. Frameworks that encompass a global scope tend to emphasize comprehensiveness, featuring numerous indicators, with a majority being internationally recognized indicators. Furthermore, frameworks in the smart city domain predominantly focus on research within single cities, specific regions, or individual domains. Regarding the disclosure of research methods and data, nearly all frameworks choose not to divulge specific details. This lack of transparency hampers the reproducibility of current research outcomes by other scholars or organizations, thereby impeding discussions and knowledge dissemination among researchers. Our hope is that this study can assist scholars and consultancy bodies in analyzing results as reference points to select their appropriate reference frameworks, tailored to specific urban development objectives. Policymakers can use the comparisons across different principles to identify the proposed indices that best align with their goals as references for future policy development.

This paper is structured into four sections. In the following section, a brief literature review of the latest smart city frameworks is presented. The methods used in this paper are illustrated in Section 2. Section 3 presents the results and analysis of the smart city frameworks. Finally, the discussion and concluding remarks are delivered in the last section.

## 2. Methods

### 2.1. Framework Selection Method

The content analysis method is used in this article to investigate the construction of smart city evaluation frameworks. Using this method, the details of smart city evaluation frameworks could be quantified and analyzed thoroughly.

Firstly, the selection of keywords is based on the commonly used vocabulary in the field of smart cities. Due to the ambiguity in the definition of smart cities, various terms such as digital cities, information cities, and sustainable cities have emerged as alternatives. However, this study focuses on comprehensive evaluations of smart cities, leading to the exclusion of content strongly implying a specific domain. Furthermore, through a literature review, it was found that there is diversity in the definition of frameworks, including terms such as model, index, framework, and assessment tool.

Hence, three main keywords are chosen: smart city index, smart city framework, and smart city assessment tools. Google Scholar and Web of Science are chosen as the search

engines since they are the largest research search engines. The filtering criteria are listed as follows.

- Relevant pieces of literature from 2010 to 2023 were screened.
- The objective of the literature should be a comprehensive assessment of smart cities. That is, the assessment should cover all areas rather than a single area. Thus, research focusing on 'smart cities 'is considered while research that only covers one field in the smart city is not considered.
- The literature must include one complete framework, such as the indicators selected, the methods adopted, etc. Literature on conceptual revision and theoretical guidance types is not counted here.
- The evaluation system is targeted at the urban scale, and its subject is the performance of urban development.

After carefully reading, 33 smart city evaluation models were finally selected for analysis and comparison. The models for analysis and comparison in this paper are listed as follows (Table 1).

**Table 1.** List of selected models in the article.

| Number | General Description |
| --- | --- |
| 1 | This index aims to provide a tool for measuring the readiness or receptiveness of city-level ecosystems for both digital startups, as well as scale-ups in Europe [15]. |
| 2 | A smart city framework for evaluating smart cities in China [16]. |
| 3 | This report describes the selection of indicators for assessing smart city projects and the corresponding indicators on the city level [17]. |
| 4 | The project evaluates medium-sized European cities and their development prospects [18]. |
| 5 | A framework with good integration, versatility, practicality, and scalability is proposed, and 17 Chinese smart cities are rated [19]. |
| 6 | The index ranks cities around the world and captures their strengths, weaknesses, and challenges in a changing world [20]. |
| 7 | Ranking smart cities in India using a taxicab distance-based approach [21]. |
| 8 | The index constructs a performance indicator (KPI) for cities from sustainable smart cities and provides a way to collect key data or information [22]. |
| 9 | This paper presents a smart city framework for implementing the concept of "smart" sustainable cities, which includes the main objectives and sub-objectives of four components: "economy and management", "quality of life", "environment", "innovation" [23]. |
| 10 | The HSE Global Urban Innovation Index (HSE GCII) presents a new way to measure the innovative attractiveness of global metropolises [24]. |
| 11 | This international standard defines and establishes definitions and methodologies for a set of indicators for smart cities [25]. |
| 12 | This index provides key performance indicators (KPIs) for smart sustainable cities and general principles for selecting KPIs to help cities achieve Sustainable Development Goals (SDGS) [26]. |
| 13 | The study uses FSE to model eight indicators for each of the six dimensions to determine the overall intelligence/intelligence level of the development index for cities in developing countries [27]. |
| 14 | This paper aims to provide an in-depth analysis of the interrelationships between the components of smart cities that connect the cornerstones of the triple helix structure [28]. |

**Table 1.** *Cont.*

| Number | General Description |
|---|---|
| 15 | This index evaluates cities as they relate to what they consider 10 key dimensions: governance, urban planning, public management, technology, environment, international outreach, social cohesion, mobility and transport, human capital, and economy [29]. |
| 16 | The goal of the index is to link a city's environmental performance to globally recognized limits [30]. |
| 17 | An evaluation model of smart city is established by using the fuzzy analytic hierarchy process method and questionnaire survey results to generate the index weight, carried out in 29 cities in China [31]. |
| 18 | A framework for the Smart Cities Environmental Sustainability Index (SCESI) was defined and evaluated to guide investment and monitor the progressive environmental development of Indian cities [32]. |
| 19 | This publication presents research findings and scientific work that advance the development and progression of the smart city and community measurement methodology [33]. |
| 20 | This article proposes an application-oriented cloud computing platform architecture, which can improve the evaluation results and maximize the capacity of smart cities [34]. |
| 21 | This study aims to offer a holistic assessment framework for smart city projects, which includes smart city dimensions and application areas [35]. |
| 22 | This index measures the comprehensiveness and ambition of urban centers against the key ingredients of a smart city [36]. |
| 23 | The article combines the hierarchical structure of areas and indicators used in smart city assessment with the concept of multidimensional assessment of facilities using the TOPSIS method [37]. |
| 24 | The index, in the form of a voucher, ranks cities around the world in 10 categories, including technology and facilities [38]. |
| 25 | In this paper, the TOPSIS method is applied to evaluate the urban agglomeration of Beijing, Tianjin, and Tangshan from seven aspects [39]. |
| 26 | This study proposed the construction of an index to objectively measure the degree of smartness in urban cities in six domain areas [40]. |
| 27 | The Smart City Framework proposed in this paper describes a process that will help key stakeholders and city/community participants understand cities' operations, city objectives and stakeholder roles, and the role of ICT within physical city assets [41]. |
| 28 | This study introduces dynamic indicators to evaluate smart cities to assess the rate of progress or regression of the category [42]. |
| 29 | The purpose of this paper is to create an information-fuzzy risk assessment model to support the municipal administration in the urban security model [43]. |
| 30 | This paper proposes an acceptance model for smart city services, which provides a basis for evaluating citizens' interaction level with smart city services based on the technology acceptance model [44]. |
| 31 | The purpose of this study is to evaluate the implementation results of Salatiga smart city by evaluating the various dimensions of Salatiga smart city [45]. |
| 32 | This paper uses six effective Multi-Attribute Decision-Making (MADM) methods to develop an intelligent assessment framework and compares and evaluates five of the most important cities in Iran [46]. |
| 33 | This paper develops a new index system, involving three dimensions of digital infrastructure, smart life, and digital economy [47]. |

The selected framework for this study comprises two types of content. One type includes smart city frameworks published globally by large organizations, such as ISO:37122 [25], which is a framework published by an international organization. The other type consists of scholarly published papers. In total, this research reviewed 33 frameworks, with 11 being published by large organizations and 22 being scholarly papers.

### 2.2. Framework Evaluation Criteria

Based on the preceding research on smart city evaluation and the summary of current frameworks, this paper will compare the models from seven aspects: generalizability, comprehensiveness, availability, flexibility, scientific rigor, transparency, and interpretability. These seven criteria encompass indicator construction, calculation methods, data utilization, result analysis, and application scope of the frameworks. The following sections will provide explanations for each of the seven criteria.

Generalizability: in the construction of frameworks for smart cities, many studies have emphasized the comprehensiveness of indicators, resulting in the overlook of indicators' generality. The indicator systems used for evaluating the development of smart cities are often intricate and extensive. Scholars aim to utilize these complex systems to assess various domains and details of the target cities. However, just as scholars are concerned about overfitting when building models, decision-makers should also take note that the construction of these complex frameworks might lead to overfitting to the target city. In other words, the selection of indicators is closely bound by the data and characteristics of the target city, and the starting point for the construction of evaluation systems is constrained by the types and amount of data available for the target city. Therefore, it is crucial to discuss the generality of indicators right from the outset of framework construction. This generality does not only refer to the ability of indicators to transfer when the target city changes, but also implies that indicators should not require significant changes over time for the same target city. Admittedly, the development of smart cities exhibits substantial variations across different stages. However, framework designers are not aiming for drastic changes in the framework within short time periods. This underscores the need to enhance the generality of framework indicators.

Comprehensiveness: Coucelis points out that the complexity of cities has led to various possible approaches to studying them [48]. As a result, the essence of cities has gained a richer core, encompassing fields such as urban economics, urban sociology, urban history, urban geography, urban ecology, urban transportation, urban health, urban anthropology, urban planning, and even urban informatics. Despite "digital cities" or "ICT cities" being considered the core of smart cities and IT industries indeed having transformed many urban areas economically, socially, and spatially [49–51], we still lack the capacity to establish a common language among citizens, urban workers, mayors, and the private sector [52]. Additionally, terms related to information and communication technology, or digital mechanisms tend to downplay certain potential urban issues and the inherent problems of labeling processes themselves [53]. It is a great challenge to propose a comprehensive framework considering the complexity of the definition and application areas [54]. In this context, Cohen proposed the Six Wheels of smart city, covering six major areas of interest for researchers and mayors. This model addresses not only the comprehensive content of smart cities but also the diverse user base. Coucelis believes that smart cities should be sustainable, livable, fair, innovative, and creative [48]. The essence of cities lies in serving humanity. Therefore, the concept of a smart city aims to make urban residents live a more convenient and happier life, achieving sustainable development [22]. Given this background, comprehensive frameworks are crucial for evaluating smart cities.

Furthermore, in the exploration of the core of smart cities, Cohen's Six Wheels of smart city has been widely accepted. Cohen posits that smart cities comprise six key domains (details can be found in Figure 1).

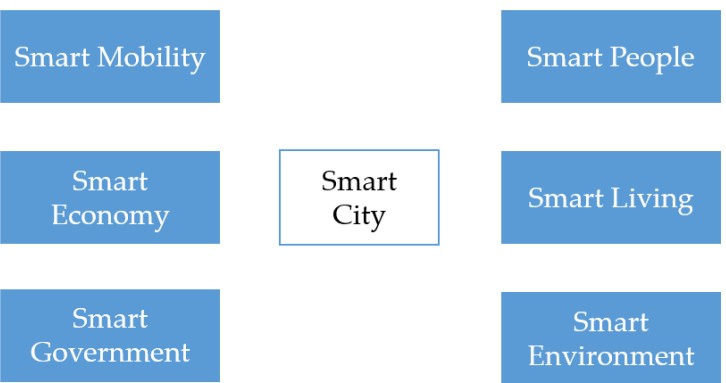

**Figure 1.** Six dimensions of smart city adapted from Cohen's six-wheel model.

- Smart People: this domain encompasses residents' physical and mental health, education, and their living environment's cultural richness, equality, and happiness.
- Smart Mobility: smart mobility involves the implementation of digital, efficient, cost-effective, safe, and environmentally friendly transportation solutions aided by ICT.
- Smart Government: smart government serves the public by operating and managing the city through various smart application systems.
- Smart Economy: smart economy emphasizes technological innovation, resource efficiency, sustainability, and high social welfare.
- Smart Environment: smart environment integrates intelligent technologies and the internet into urban environmental management and pollution control.
- Smart Living: smart living aims to enhance urban life's convenience, satisfaction, social and digital inclusion, housing conditions, and safety.

Since these frameworks are tailored for specific objectives, they are not directly comparable to users or stakeholders. However, this does not imply a lack of importance for comprehensive users or stakeholders. On the contrary, identifying the purpose and user base of smart city assessment frameworks is vital for designers. This text refrains from statistical comparisons since each framework has its intended audience.

In this study, not only the breadth of indicator construction in these smart city evaluation frameworks was investigated, but also the equilibrium among indicators from different domains was analyzed. The concept of sustainable development assumes a clear balance among social, economic, and environmental development goals [55]. Therefore, a comprehensive assessment should not only consider the scope covered by the framework but also pay attention to the balance among indicators from different domains. When designing a smart city evaluation framework, it is essential to ensure that indicators from different domains can complement each other, facilitating a holistic assessment of the city's development across various aspects. Such balance aids in avoiding overemphasizing a particular domain while neglecting other crucial domains, thus enabling a more comprehensive evaluation of the overall development level of the smart city.

Availability: due to the versatility of the same data for multiple purposes, data availability not only provides the potential for replicating frameworks to researchers in smart city evaluation but also serves as a reservoir for a wide range of studies in the field of smart cities. In other words, this approach not only enhances traceability, thereby improving accountability for research outcomes [56], but also offers a reference for data acquisition in various studies related to smart cities. Despite data accessibility being globally constrained by factors such as societal considerations and the interests of stakeholders, as evidenced by challenges faced in achieving the data and data collection systems for the United Nations 2030 Agenda, the implementation of smart cities through technology and big data capabilities can help overcome these challenges [57]. Regulations and standards related to data protection and network security, as critical components of smart city governance, are considered essential for guiding the development and scalability of digital infrastructure

and services [58]. Therefore, data availability should still be recognized as an indispensable feature of influential smart city frameworks.

The concept of availability being discussed here goes beyond just the data available to the designers of the framework; it extends to the data available to anyone who engages with or uses the framework. Data availability is checked by three dimensions of a framework: (a) whether the framework exposes the used database or, (b) whether the framework exposes the detailed data source or, (c) whether the data could be accessed after permissions.

Flexibility: ideally, a smart city framework should not be influenced significantly by location, scale, or time, or should minimally be influenced by contextual factors. On one hand, a flexible framework can avoid frequent changes, reducing costs for organizers and researchers. Moreover, the flexibility of indicator sets assists designers in adjusting assessment schemes according to specific urban needs and priorities [4]. Furthermore, the development of smart cities emphasizes robustness and sustainability. The flexibility of a framework ensures the sustainability of the evaluation system. On the other hand, flexibility, generality, and comprehensiveness are interdependent. When generality and comprehensiveness are high, the breadth and depth of the model have already reached a considerable level. In such cases, minor adjustments in time and location would not significantly affect the model. In contrast, when a model is sufficiently flexible, and unrestricted by location, year, or scale, the model's indicators must exhibit universality and comprehensiveness. Although it is hard to achieve a totally flexible framework due since those models come from different regions of the world and were built based on distinct components [7], flexibility is still an important consideration before constructing frameworks.

Scientific: the scientific rigor of a framework is a primary concern for all researchers. The concept of "scientific rigor" encompasses various aspects, including indicator selection, weighting methods, and framework structure choices. "Scientific thinking" in this context is seen as goal-oriented behavior (practice) and its product, as well as the nature of the question posed based on that goal and appropriate means of achieving it [59]. Hence, we can judge scientific rigor based on the alignment of goals and means. However, indicator selection and structure choice to some extents are subjective decisions, making it challenging to determine their scientific nature. Since weighting methods often involve precise mathematical and statistical techniques, this paper primarily evaluates the scientific rigor based on the weighting methods. The selected model will be examined using the following process outlined in Figure 2.

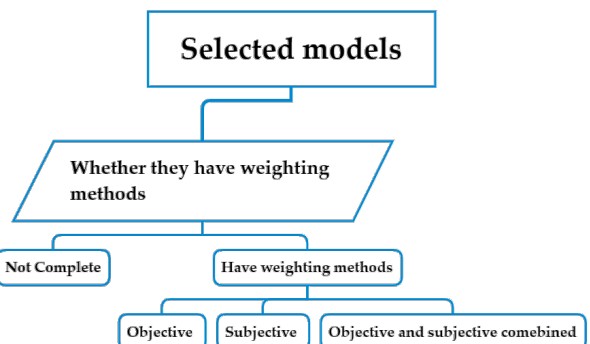

**Figure 2.** Checking process framework for the criterion 'Scientific'.

Transparency: many researchers or organizations often overlook the transparency of a framework. The transparency of a framework frequently ensures its replicability and provides opportunities for receiving more suggestions at every step of its construction. Transparency implies that each component is scientifically and thoroughly described. This encompasses the entire process of framework development, including indicator selection, data sources or collection methods, weighting methods and final weights determination, calculations, data processing, and more. These processes span multiple interdisciplinary

fields, including indicator selection related to social sciences, data quality assessment associated with data analysis, and weighting methods and data aggregation related to statistics. The transparency of framework design enables the incorporation of insights from scholars and practitioners from various fields, serving as a baseline for further refinement and progress. Transparency is crucial for scientific research, as it determines the practicality and impact of scientific endeavors. In essence, it is transparency that grants the frameworks we construct their replicability, survivability, and scalability.

Interpretability: the interpretability of the results is another often overlooked aspect when designing a smart city assessment framework. Typically, most individuals consider the structure we employ, the final weights, and the ultimate ranking results or assessment scores as the endpoint. However, what this framework can contribute to our cities or society is what truly matters to users. Users of a smart city assessment framework expect the results to uncover shortcomings and provide insights for future development directions. Moreover, the outcomes of smart city assessments play a critical role in guiding the exploration of development stages and priorities. Hence, it is essential to perform an analysis and interpretation of the assessment framework's results. Defining whether a model has a social impact or successfully explains its results is indeed a challenging task. In this study, the presence of an analysis section within the selected frameworks' documentation was analyzed and taken as evidence of the framework's interpretability. These analysis sections may vary due to the framework's purpose and nature. For instance, the analysis of globally applicable indicators, e.g., the Power City Index, involves summarizing insights from various domains, while the IMD Smart City Index offers brief analyses of each selected city. Other frameworks designed in the articles might incorporate analyses of smart city development technologies or in-depth analyses of sample cities. These variations do not hinder the exploration and pursuit of result interpretability. Enhancing the interpretability of the results enhances the readability of smart city assessment frameworks, lowers the reading threshold of such research materials, and broadens their potential for development.

## 3. Results

As mentioned above, an in-depth investigation has been made to analyze the commonality and differences among the selected 33 smart city index models. Seven criteria have been discussed here concerning the construction of a smart city framework. To avoid the problem of incomplete text display of the image axis, the following analysis of individual frames will replace the original name of the frame with the article number.

### 3.1. Generalizability

To assess the generalizability of the indicators within the chosen smart city frameworks, it is imperative to initially compile a comprehensive list of these indicators and subsequently categorize them based on their respective definitions and thematic scopes. In this context, the indicators are categorized into two distinct groups: those of international significance and those tailored to local contexts. The criteria employed for this classification are as delineated below:

- The index is of an international research nature, characterized by a standardized definition and computation methodology, akin to metrics such as GDP, Gini Index, and birth rate, or;
- A comprehensive worldwide index disseminated by international institutions or research entities, or;
- Indicators that can be mutually converted due to disparities in definition and calculation arising from national and regional differences.

A total of 1259 indicators of the selected models (seven models are excluded due to lacking details of indicators) have been investigated. The analysis in Figure 3 shows that only 37% of indicators are international and the others could only be applied locally.

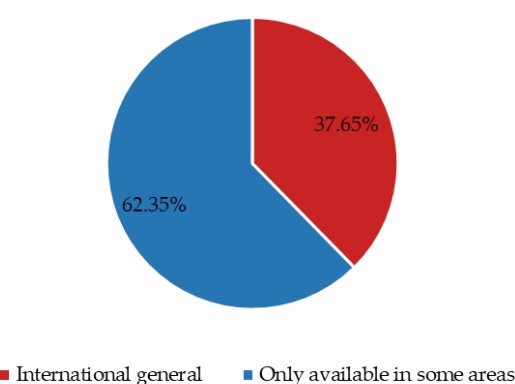

■ International general   ■ Only available in some areas

**Figure 3.** Analysis result pertaining to the criteria 'Generalizability'.

The results depicted in Figure 4 confirm this assertion, that a limited number of frameworks consider the universality of indicators. Most frameworks did not exhibit more than 50% common international indicators, or even none in some cases. As a result, these models lack repeatability across different cities or even within the same city over subsequent years, which could hinder the progression of urban development. Additionally, the more distinctive the unique attributes specified by a framework for a city are, the more it might be detrimental to the expansion of research scope and depth. However, we also observed that with the passage of time, an increasing number of framework designers are opting for more universally applicable indicators. This indicates a growing awareness among designers towards the issue of indicator universality.

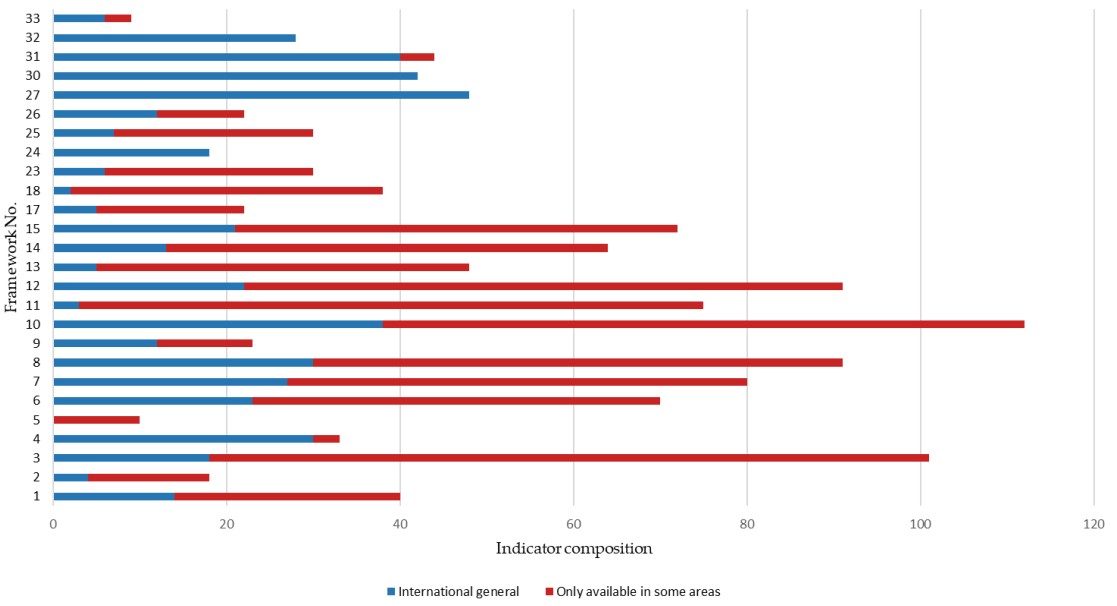

■ International general   ■ Only available in some areas

**Figure 4.** Analysis result pertaining to the criteria 'Generalizability' in the selected frameworks.

### 3.2. Comprehensiveness

In this article, the comprehensive criterion is checked based on Cohen's six-wheel model for smart cities.

As demonstrated by the analysis results (Figure 5), a discernible lack of equilibrium exists among indicators across various dimensions within the customary smart city framework. Among the 32 chosen frameworks (Framework 30 discusses all the subjective issues surrounding smart service, which cannot be divided into six dimensions, and thus will not be discussed here), the themes of smart living, smart environment, and smart economy emerge as particularly prominent focal points. Researchers exhibit a pronounced inclination towards these three dimensions, dedicating heightened attention to their exploration.

However, these dimensions are typically overseen by dedicated departments within cities. For instance, the management of the smart environment falls under the purview of the city's environmental department. Consequently, it is highly plausible that a substantial volume of data or information can be readily obtained. This, in turn, accounts for the comparatively substantial representation of such indicators within the evaluation framework.

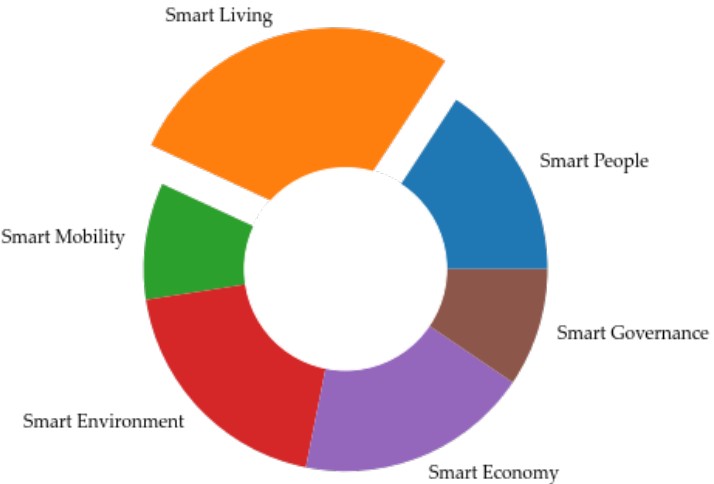

**Figure 5.** Analysis result pertaining to the criteria 'Comprehensive'.

For more comprehensive insights, the allocation of indicators in each of the chosen frameworks is explored in Figure 6. Broadly speaking, the distribution of indicators in most models is similar. Evidently, a limited number of frameworks manage to attain equilibrium among indicators across all six dimensions, with challenges observed among the larger-scale evaluation frameworks. Notably, the categories of smart environment, smart economy, and smart living constitute the three most substantial segments, albeit with slight adjustments depending on the specific subject or domain of inquiry.

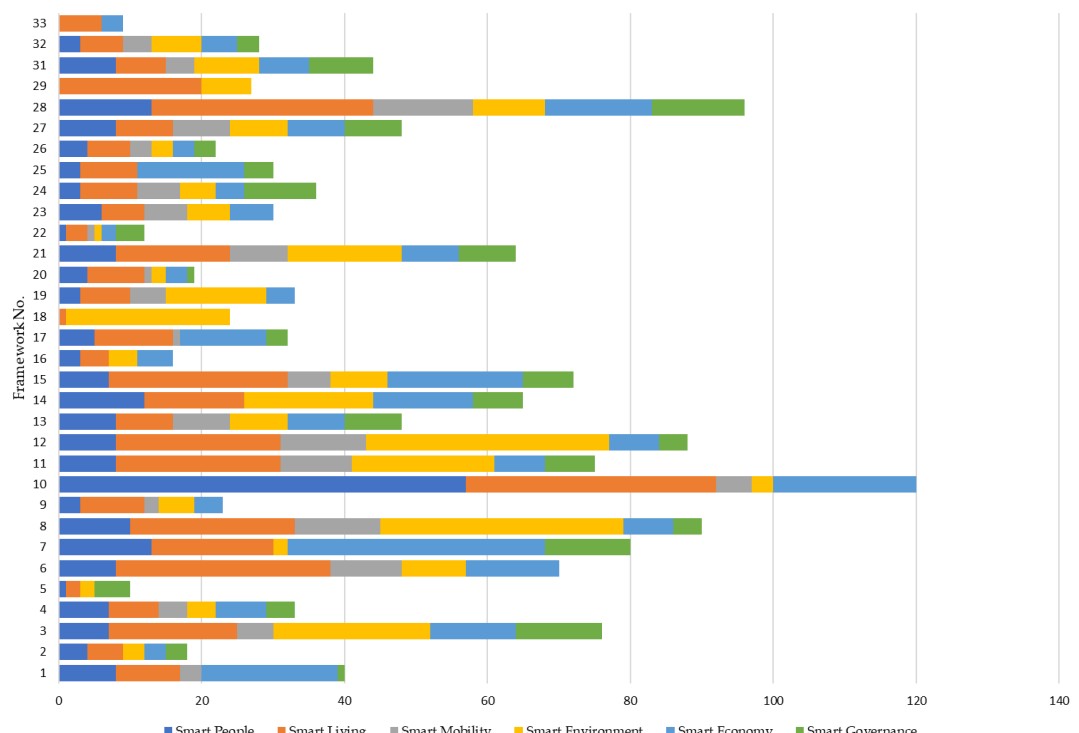

**Figure 6.** Analysis result pertaining to the criteria 'Comprehensive' in the selected frameworks.

This raises a pertinent question: should designers strive for a balanced distribution of assessment indicators across diverse domains to ensure impartiality, or should designers tailor the ratios to align with the actual circumstances and research objectives? Determining the correct approach remains challenging, yet a critical principle should be upheld when formulating smart city frameworks to ensure comprehensiveness. If the indicator balance of the framework cannot be guaranteed, this problem should be dealt with during weighting. This practice aims to forestall any undue bias toward cities that excel or lag in specific aspects during evaluations. In other words, a city that excels significantly in one dimension but performs less commendably in others could garner inflated scores due to a higher number of indicators in that dimension, resulting in a potentially inflated overall score. Conversely, cities with deficiencies in a particular facet may similarly receive disproportionately lower scores, resulting in compromising the framework's equity and logical coherence.

### 3.3. Availability

Among the chosen frameworks, a mere 30% furnished information regarding data sources (refer to Figure 7). A scant three frameworks out of those disclosing data sources exclusively relied on public data (as depicted in Figure 8). Most frameworks exhibit considerable vagueness with regard to data sources, often resorting to terms such as 'may obtain from…'. This prevailing scenario significantly undermines the framework's influence and practicality.

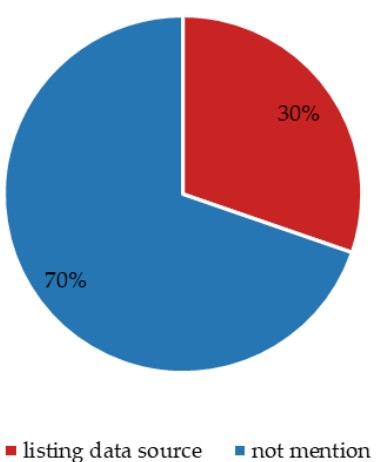

**Figure 7.** Analysis result pertaining to the criteria 'Availability'.

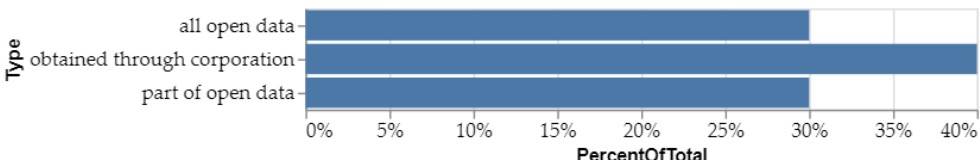

**Figure 8.** Analysis result pertaining to the criteria 'Availability' in the selected frameworks.

This predicament stems from a combination of technical and ethical challenges. Firstly, the adoption of electronic open data remains limited, primarily prevalent in well-developed nations and regions possessing comparatively established electronic open data repositories. Nevertheless, even in these advanced urban centers, their public data repositories are continually undergoing structural transformations, and the cadence of updates remains variable. Conversely, the constraint of privacy data protection poses an additional barrier; not all the data requisite for evaluation can be openly divulged. This predicament is particularly pronounced for models developed through government or corporate funding initiatives.

Nonetheless, there is an undeniable trajectory toward enhanced data availability as technology continues to evolve. The advancements in information obfuscation techniques and data acquisition technologies, coupled with the gradual maturation of urban electronic open data platforms, are poised to amplify the reservoir of open data. Consequently, factoring in data availability during the formulation of smart city evaluation frameworks assumes significance as a vital reference point for models aiming to ensure sustainability.

### 3.4. Flexibility

In the context of differences in study scale and geographical coverage (depicted in Figure 9), it is evident that alterations in the study area are significantly influenced by the identity of the framework's creators. As illustrated in the chart below, ten frameworks emerge as products of global research endeavors, crafted through collaborative efforts involving academic consortia within prominent laboratories, collaborative initiatives between research institutes and large corporations, or group undertakings facilitated by international organizations. Conversely, frameworks centered around a single city or nation predominantly originate from the purview of university professors, which are constrained by the inherent limitations posed by the available human and material resources. Notably, regional investigations presently concentrate primarily on Europe, owing to the richness of accessible public data sets and the established practice of data updates at fixed intervals within the European Union. Frameworks that abstain from divulging their specific research domain generally pertain to scholarly explorations of novel model structures or weighting methodologies, wherein practical implementation is not pursued.

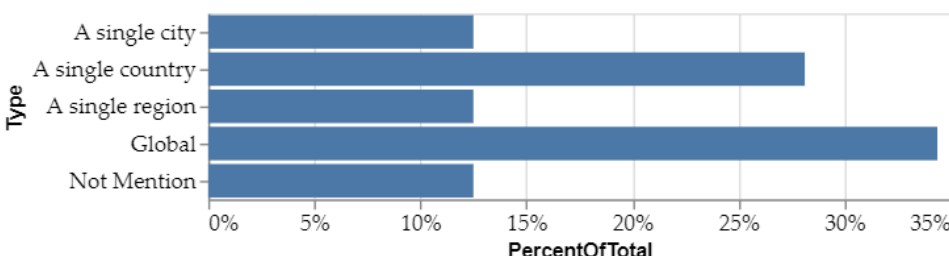

**Figure 9.** Analysis result pertaining to the criteria 'Flexibility'.

Within the spectrum of temporal variations in study durations, a limited number of frameworks have succeeded in sustaining continuous updates and furnishing annual reports. However, it remains undeniable that nearly all models that have consistently adhered to prolonged, recurrent updates and reporting hold substantial international influence.

### 3.5. Scientific

Of the selected frameworks, Figure 10 highlights that 64% of the frameworks exhibit a significant weighting approach, while the remaining 36% do not. Over time, the newer the framework, the more emphasis on the definition of weighting methods. This reflects a shift in research focus. More and more researchers have begun to pay attention to the scientific nature of weighting methods.

Regarding the selection of frame weighting methods (as shown in Figure 11), 13 adopted objective methods (62%), seven adopted subjective methods (33%), and one adopted a mix of subjective and objective methods (5%). Compared to subjective methods, objective methods are indeed favored for their versatility and efficiency. However, it is worth noting that objective methods often exhibit a tight alignment with data characteristics and are sometimes overly dominated by data distribution patterns and characteristics, resulting in results contrary to the designer's intent.

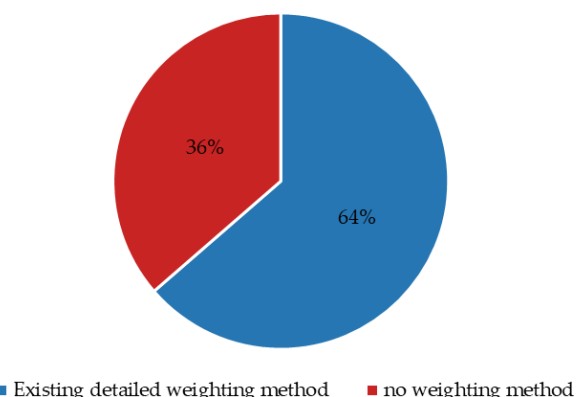

**Figure 10.** Analysis result pertaining to the criteria 'Scientific'.

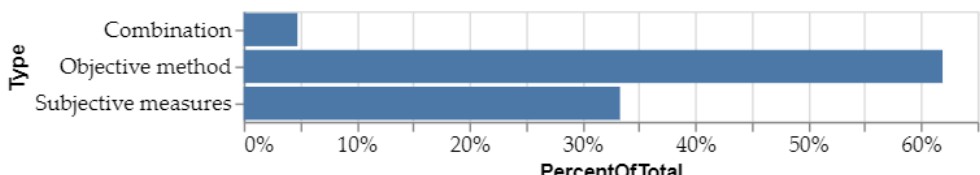

**Figure 11.** Analysis result pertaining to the criteria 'Scientific' of specified weighting methods.

Although subjective methods possess the capacity to amalgamate expert opinions, they tend to be time-intensive and resource-demanding. On the other hand, the amalgamation of subjective and objective methods capitalizes on the merits of both, rendering the design process more scientific and efficacious. However, this approach also integrates the drawbacks of both techniques, resulting in heightened temporal and resource consumption, as well as augmenting the intricacy of calculation method design.

### *3.6. Transparency of Process*

In scrutinizing the transparency of the chosen frameworks, the primary focus lies in assessing whether these models encompass comprehensive elucidations or distinct documentation for each facet of their construction.

An analysis of selected frameworks (Figure 12) sheds light on the transparency deficiencies observed during the design of smart city frameworks. Of the selected frameworks, 33% lacked information about the calculation method and the relevant weights for each indicator. The other 30% of the selected frameworks only provide a concise rationale for the methods employed, thereby obscuring the process and usage, 37% of the selected frameworks have a comprehensive description of the principle of the calculation method, the procedure steps, and the derivation of the results. It is worth noting that frameworks willing to provide detailed computational procedures and principles appear mainly in academic articles. A common feature of these frameworks is that they often lack practical validation or are limited to validation within selected cities in a single country.

Based on the analysis, while transparency holds substantial importance in enhancing the influence of the smart city framework and facilitating cross-framework reference and exchange, the comprehensive disclosure of intricate framework specifics remains a formidable undertaking. The intricacies of framework disclosure are notably constrained by various factors, primarily stemming from the diverse investors associated with the smart city framework. The exhaustive elucidation of model intricacies encounters obstacles on multiple fronts. Academics, for instance, might exhibit reticence in divulging intricate details concerning weighting and scoring methods, particularly in cases involving innovative approaches. Similarly, enterprises may be disinclined to unveil exhaustive data collection and processing specifics due to implications related to user privacy, corporate confidentiality, or other vested interests.

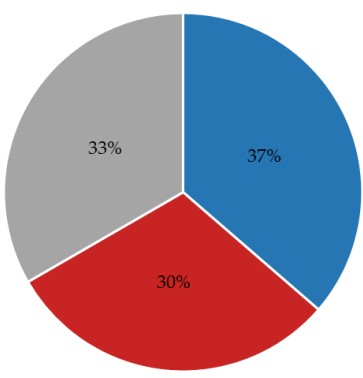

- Transparent calculation process
- Including simple calculation methods but not disclosing the calculation process
- Not mentiond

**Figure 12.** Analysis result pertaining to the criteria 'Transparency'.

In summary, the journey towards achieving transparency in framework design and results remains a significant ongoing endeavor. Nonetheless, this should not deter the integration of this principle into the framework design process. Rather, it underscores the importance of making judicious modifications in result disclosure. This approach aims to offer insights and recommendations for forthcoming model designs, thereby fostering inspiration and guidance in the pursuit of improved practices.

*3.7. Interpretability of Results*

The analysis shown in Figure 13 shows that, in the selected framework, 55% provided an analysis of the final score or city ranking. In contrast, 15% only provide final rankings or city scores, and 30% do not provide any results. The frameworks that provide the analysis of the results are usually designed or implemented by international organizations, large corporations, or academic institutions working with businesses, or the theoretical practice of academics. These analyses mainly focus on understanding the distribution of scores in various areas within the city or the difference analysis of city performance and infer smart city development trends and future hot-spots.

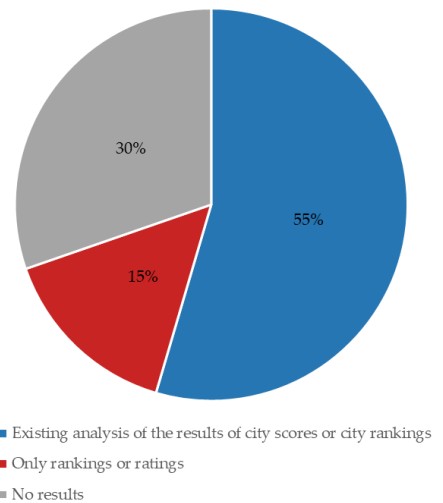

- Existing analysis of the results of city scores or city rankings
- Only rankings or ratings
- No results

**Figure 13.** Analysis result pertaining to the criteria 'Interpretability'.

Furthermore, a segment of the analysis focuses on individual cities, encompassing comprehensive evaluations of city facets to formulate comprehensive city reports, a practice commonly favored by urban planners and developers. Frameworks crafted through academic-government or academic-organization collaborations often yield reports for

each city within the study area. In the case of international organizations or corporations, the final analysis frequently leans towards rankings and projections of overarching trends. For enterprises, city ranking shifts often yield impactful news and generate significant attention—an outcome that aligns with their needs and expectations.

In essence, the divergent final analysis outcomes across distinct evaluation frameworks stem from variances in objectives. Nevertheless, the absence of result analysis significantly undermines the completeness of an evaluation framework. Within the context of evaluating smart cities, the analysis and presentation of results constitute a pivotal aspect in deciphering the fundamental essence of the evaluation framework. Typically, the comprehension of various facets within the framework's design, coupled with anticipations for the future trajectory of smart city development, becomes unveiled through the analysis of results.

During the initial incorporation of indicators, each individual metric affords decision-makers insights into the designer's perspective on smart cities through the lens of that specific indicator. After the computation, the results encapsulate the real-world application efficacy of the composite framework within an urban context. This outcome unveils the dynamic interplay between conceptual design and practical implementation, reflecting the vibrancy that emerges through the collision of these realms.

## 4. Discussion and Conclusions

This study analyzed the characteristics of 33 smart city frameworks from seven aspects. The research findings revealed that early smart city assessment frameworks had lower universality and higher limitations in indicator selection, while recent frameworks exhibited higher universality in indicator choices, leading to increased portability of the frameworks. In terms of the comprehensiveness of the framework system, the areas of Smart Living and Smart Environment emerged as favored domains among researchers, followed by Smart Economy and Smart People. However, despite the gradual refinement of definitions for smart cities in recent years, frameworks considering holistic city evaluations have gradually encompassed more comprehensive domains. Nevertheless, there is still room for improvement in achieving the balance between different domain indicators. Regarding data availability, there has not been a significant increase due to technological advancements. Generally, smart city frameworks willing to provide data sources remain scarce. Moreover, early smart city frameworks with a global or regional focus were more prevalent, but recent frameworks have increasingly focused on individual cities or countries. Over time, more smart city frameworks are willing to disclose specific experimental methods and calculation processes, which has greatly facilitated the further development of weighting and aggregation methods. However, in terms of result analysis, in-depth and thorough analyses are predominantly found in academic papers, while larger-scale smart city frameworks tend to focus on descriptive phenomena.

Drawing insights from evaluation materials and contemporary smart city frameworks, this study offers comparative principles for smart city frameworks as reference points. These principles provide macro-level design guidance to make framework construction more scientific and purpose driven. Scholars and consultancy agencies can use the analyzed results of this study to select other frameworks that can serve as references for their specific city development goals, thereby choosing or designing evaluation frameworks that are better suited to their needs. Policymakers can use these principles as a reference to determine the suitability of proposed indices for their goals.

Currently, many cities around the world are in the early or initial stages of smart city development. There is still a lack of consensus on a clear definition and developmental trajectory for smart cities globally. The goal is to establish a forward-looking, sustainable, transferable, and high-potential smart city assessment framework. This initiative aims to provide a blueprint for the design and direction of smart city frameworks for developing cities or those in the early stages of smart city adoption. Meanwhile, cities with research outcomes or mature systems can draw inspiration from this framework to assess and improve existing frameworks. This collaborative approach seeks to collectively guide smart

cities toward a more comprehensive, transferable, adaptive, and impactful trajectory of development.

This study is subject to certain limitations that warrant consideration. Firstly, the ambiguity in defining the concept of a smart city introduces a challenge. Despite our earnest efforts to comprehensively cover diverse frameworks by referencing pertinent antecedent literature and selecting relevant keywords, the multifaceted nature of smart city concepts precludes an absolute guarantee of encompassing all potential frameworks.

Secondly, this study focuses on establishing design principles for a comprehensive smart city evaluation framework. However, the applicability of these principles as references for domain-specific evaluation frameworks may be constrained. While these design principles hold promise in offering general guidance for evaluation frameworks across different domains, their practical application and relevance within specific sectors necessitate further exploration.

Hence, although this study furnishes crucial guiding principles for the design of smart city evaluation frameworks, it remains imperative to account for the limitations during real-world implementation. Adaptable adjustments and applications, contextualized by specific scenarios, are essential. Subsequent research endeavors have the potential to delve deeper into these principles and explore additional smart city evaluation frameworks tailored to diverse contexts and developmental stages.

**Author Contributions:** Conceptualization, Fan Shi and Wenzhong Shi; methodology, Fan Shi and Wenzhong Shi; formal analysis, Fan Shi; writing—review and editing, Fan Shi and Wenzhong Shi; visualization, Fan Shi; supervision, Wenzhong Shi; project administration, Wenzhong Shi; funding acquisition, Wenzhong Shi. All authors have read and agreed to the published version of the manuscript.

**Funding:** This work was supported by Urban Informatics for Smart Cities, The Hong Kong Polytechnic University under Grant [1-ZVN6], and the Otto Poon Charitable Foundation Smart Cities Research Institute, Hong Kong Polytechnic University under Grant [CD03].

**Data Availability Statement:** No public data used in this literature review.

**Conflicts of Interest:** The authors declare no conflict of interest.

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
