# Peer review of "A Critical Review of Smart City Frameworks: New Criteria to Consider When Building Smart City Framework"

_ijgi, doi:10.3390/ijgi12090364_

Round 1

Reviewer 1 Report

The paper has an interesting and actual topic and the authors are going in deep regarding smart city problematics, taking into account a list of selected models. The analysis could be considered very useful for those who are interested on state of the art in smart city topic. The study proposes new design principles for smart city models, however the conclusions section must be improved. An important shortcoming could be that the most of references are papers that had been written before 2020. The authors should take into consideration that the smart city modeling concept increased a lot as methods and technology after 2020.

Reviewer 2 Report

This study selected 26 smart city evaluation models to analyze and compare them from the 7 aspects of generalizability, comprehensiveness, availability, flexibility, scientific, transparency, and interpretability. Though this work is interesting, it needs major revision. Some of the comments are listed below:

1. Line 13-16: ‘This study selected 26 evaluation frameworks proposed in recent years or large-scale research projects that have lasted for many years and analyzed and compared them from the 7 aspects of generalizability, comprehensiveness, availability, flexibility, scientific, transparency, and interpretability.’ Why analyze 26 evaluation frameworks? Why compare these frameworks from the 7 aspects? The previously mentioned context is not sufficient to support the aim of this study. 

2. Line 46-56: The author cites sufficient literature to summarize the gap in the current research field. However, it should be noted that it is not possible to simply list the content of the literature, but to increase the relationship between each other, thereby leading to the importance of the research problem.

3. Please clarify the research object of this paper. With regard to the smart city frameworks in the title, there are various forms of expression in this paper, such as ‘26 evaluation frameworks proposed in recent years or large-scale research projects’, ‘26 smart city evaluation models’, ‘26 smart city index models’, and ‘smart city models’. 

4. Line 131-133: Figure 2 shows the results of the analysis for only 18 smart city frameworks. Why not show the analysis results of 26 frameworks, even though the text mentions that six frameworks are missing detailed information? In addition, why is the smart city framework called ‘IMD’ not content in the Figure? 

5. Line 161-162: ‘In this article, the comprehensive criterion is checked based on Cohen’s six-wheel model for smart cities.’ Cohen’s six-wheel model can be described in more detail to give the reader a better understanding.

6. Line 167-168: ‘As the analysis shows (Figure 3), the indicators in different fields are not balanced in the usual smart city framework.’ This sentence actually corresponds to ‘Figure 2 Figure 2 Analysis result concerned with the criterion ‘Comprehensive’ in total’. There are many such issues in this paper, please carefully check. In addition, what rules were used to categorize the indicators in the 26 smart city frameworks into different dimensions? Furthermore, there are two ‘Figure 2’ and two ‘Figure 8’ in this paper. 

7. There are also many formatting issues in this paper that do not meet the requirements of the journal. First, please carefully check the punctuation in this paper to ensure that it is in the format of Palatino Linotype and remove redundant spaces and punctuation. Second, please check the format of titles at all levels and chart titles according to the journal requirements. Third, please check the reference format according to the reference list and citations style guide provided by the journal. Fourth, ensure the same format (size, font) for charts of the same type in this paper, such as Figure 1, Figure 4, etc. In addition, the coordinate content of many bar charts in this paper is not fully displayed. 

8. There are still grammar and spelling issues in this paper, please check and polish it.

There are still grammar and spelling issues in this paper.  Extensive editing of English language required.

Reviewer 3 Report

Overall, I have an issue with a structure of the paper and clarity of its goals, and especially the methodology of papers ref. 8-33 choosing. Maybe the research has been done in a manner of quality, but I have a problem with of its overall presentation.

1. Lines 75-78 statements are not in accordance with the paper structure, Methods are in chapter 2 and results are in chapter 3.

2. Line 87, why these words and not some others?

3. Lines for instance 183, 184 instead of “we” it should be neutral speech throughout the paper.

4. Why the papers in Table 1 are chosen and not some others? Also, citation usually goes with explaining the content of the paper and in table there are only the titles of the papers.

5. Lines 70-72, why investigating these six elements and not some others?

6. Figure numeration is not ok all over the paper.

7. For a review paper it should be much more references cited, especially since references 8-33 are not cited, but are research material. Previous research on the topic is not sufficient.

8. The questions that should be clearly raised by the paper are:

- what are its goals;

- who is it useful for;

- what questions should it answer on;

- what are the weaknesses of the research;

- what are the recommendations and implications of the research;

- how could/should futures studies improve the model etc.

Reviewer 4 Report

The objective of this paper is to critically analyze and compare the existing smart city evaluation frameworks proposed in recent years or large-scale research projects that have lasted for many years. The paper aims to identify the strengths and weaknesses of these frameworks and provide guidance for the development of future smart city evaluation models. The innovation of this study lies in its critical analysis and comparison of 26 smart city evaluation frameworks proposed in recent years or large-scale research projects that have lasted for many years. There are several comments below.

1. Provide a more detailed explanation of the seven aspects used to analyze and compare the smart city evaluation frameworks.

2. Include a section that discusses the limitations of the study and areas for future research.

3. Provide more examples of the strengths and weaknesses of the evaluated frameworks to make the analysis more concrete.

4. Include a section that discusses the practical implications of the study for policymakers, city planners, and other stakeholders involved in the development of smart cities.

5. Provide a more detailed explanation of the methodology used to select the 26 evaluation frameworks analyzed in the study.

I have no other comments on the English language.

Round 2

Reviewer 2 Report

The revised paper has made some modifications and adjustments to the comments. But there are still some problems that can be improved, some of the comments are listed below:

Line 13-16: ‘This research undertook a comprehensive analysis and comparative assessment of 33 evaluation frameworks, either recently proposed or derived from extensive, long-term research undertakings.’ What is the relationship between this sentence and the previous research background? Why this research is being conducted seems unclear.

Line 59-63: ‘"The outcome of the conducted analysis defines the smart city as 'an innovative city that uses ICTs and other means to improve quality of life, the efficiency of urban operations and services and competitiveness while ensuring that it meets the needs of present and future generations with respect to economic, social and environmental aspects'"’ Please check if the quotation marks in the entire text are in Palatino Linotype format. In addition, please check if this superimposed quotation mark is correct.

At present, the font in the figures is not uniform, it is recommended to unify the font.

There are also some formatting and punctuation issues in the paper, such as items 4 and 10 in Table 1.

Minor editing of English language required.

Reviewer 3 Report

The authors made some changes of significance. The paper is improved. My only remark is that 5. Limitations should be an integral part of conclusion and should not have a title or subtitle.

Author Response

Thank you for your suggestion. We have moved the limitation section into the end of the conclusion section.

Reviewer 4 Report

The author answered my questions well, I have no other questions, but the English needs to be touched up again by a NATIVE speaker.

English needs to be touched up again by a NATIVE speaker.

Author Response

Thank you for your suggestion. We have polished the article again.